

# Local-scale impact of wind energy farms on rare, endemic, and threatened plant species

Mihaela Urziceanu[1,2], Paulina Anastasiu[1,2], Laurentiu Rozylowicz[3] and Tatiana Eugenia Sesan[1,4]

[1] Department of Botany & Microbiology, Faculty of Biology, University of Bucharest, Bucharest, Romania
[2] Botanic Garden "D. Brandza", University of Bucharest, Bucharest, Romania
[3] Center for Environmental Research and Impact Studies, University of Bucharest, Bucharest, Romania
[4] Academy of Agricultural Sciences and Forestry, Bucharest, Romania

Corresponding author
Paulina Anastasiu,
paulina.anastasiu@bio.unibuc.ro,
anastasiup@yahoo.com

## ABSTRACT

**Background**. Wind energy farms have become a popular solution to produce green energy worldwide. Their development within protected areas has increased dramatically in the past decade, and the effects on the rare, endemic and threatened plant species (i.e., protected plant species), essential for habitat conservation and management, are little known. Only a few studies directly quantify the impacts of wind energy farms on them. Our study analyzes the impact of wind energy farms on rare, endemic, and threatened plant species in steppic habitats and their recovery potential over a ten-year period on a wind energy farm within the Dealurile Agighiolului Natura 2000 site (Dobrogea Region, SE Romania).

**Methods**. We surveyed the rare, endemic, and threatened plant species within a radius of approximately 50 m around each of the 17 wind towers during the wind farm operational phase. We selected 34 plots to allow the investigation of two types of areas: (1) a disturbed area overlapping the technological platform, where the vegetation was removed before construction, and (2) an adjacent undisturbed area. To understand the effects of the wind energy farm on the rare, endemic, and threatened plant species diversity and the differences between the disturbed and undisturbed areas, we calculated under both conditions: (1) plant species richness; (2) sample-size-based rarefaction and extrapolation with Hill numbers parameterized by species richness; (3) non-metric multidimensional scaling of Jaccard dissimilarity index; (4) functional diversity; (5) beta-diversity (including replacement and nestedness of species).

**Results**. As a result of the disturbances caused by the wind energy farm's development, we identified a sharp contrast between the diversity of rare, endemic, and threatened plants inhabiting disturbed and undisturbed areas near the wind towers. Our research showed that less than 40% of the total inventoried rare, endemic, and threatened species colonized the disturbed sites. Species turnover within undisturbed plots was higher than disturbed plots, implying that the plant community's heterogeneity was high. However, a higher richness in rare, endemic, and threatened plant species was found in the plots around the wind towers in grasslands of primary type. Sample-size-based rarefaction and extrapolation with Hill numbers by observed species richness indicated an accurate estimation of species richness in disturbed habitats, demonstrating that recovery after wind energy farm construction was incomplete after ten years of low-intensity plant restoration and conservation activities. Thus, we consider that operating activities must

be reconfigured to allow the complete recovery of the communities with rare, endemic, and threatened plant species.

# INTRODUCTION

Plants are a vital component of biodiversity and play a key role in maintaining ecosystems' ecological stability, e.g., by providing irreplaceable ecosystem services (*CBD, 2012*). Human society depends on economic growth, and the depletion of resources dramatically transforms the environment, driving species towards extinction at an unprecedented rate in human history (*Corlett, 2016*; *Johnson et al., 2017*; *Meng et al., 2019*). Plant species are exposed to higher extinction risk due to habitat loss and fragmentation, competition with invasive species, and climate change effects, yet available plant conservation initiatives are overlooked compared with wildlife conservation (*Mouillot et al., 2013*; *Corlett, 2016*; *Zambrano et al., 2019*).

The establishment of protected areas is a major tool for conserving biodiversity worldwide (*Miu et al., 2020*). Protected areas were created to safeguard biodiversity, more so of rare, endemic, and threatened plant species, but are presently facing pressures such as dense transport infrastructure, over-tourism, intensive agriculture, habitat transformation, and altered hydrological and fire regimes (*Schulze et al., 2018*). The development of infrastructure for renewable energy production is regarded as a threat to protected areas; moreover, wind energy farms can significantly impact biota where large-scale systems are developed (*Schulze et al., 2018*). Threats to biota from wind energy farm development typically include increased mortality of birds and bats, alteration of habitats and landscapes, and increased noise (*Kuvlesky Jr et al., 2007*; *Katsaprakakis, 2012*; *Gasparatos et al., 2017*; *Măntoiu et al., 2020*), while immediate and long-term effects on plant species are less studied (*Silva & Passos, 2017*; *Nita, 2019*). However, research on the impact of wind energy farms on plant species is gaining traction. Several studies have demonstrated a reduced diversity of plant species close to wind energy farms and the displacement of rare, endemic, or threatened plants by temporary ruderal and invasive species (*Fraga et al., 2008*; *Renou-Wilson & Farrell, 2009*; *Passos et al., 2013*; *Silva & Passos, 2017*; *Keehn & Feldman, 2018*). For example, *Keehn & Feldman (2018)* indicated that plant communities disturbed by wind energy farms in Southern California (USA) are less abundant and diversified in endemic plants than areas without wind energy farms. In Europe, several studies focusing on the impact of wind energy farms on mires and peat habitats suggest that the diversity of plant species is diminished in disturbed habitats and that their impact is particularly prominent on rare and threatened plant species (*Dargie, 2004*; *Fraga et al., 2008*; *Fagúndez, 2008*; *Renou-Wilson & Farrell, 2009*; *Natural England, 2010*).

Furthermore, several studies indicated that during operation, wind farms might affect the local climate (temperature and rainfall) (*Zhou et al., 2012*; *Tang et al., 2017*), resulting

in an additional stress factor for vegetation growth. *Xu et al. (2019)* observed positive ecological effects of wind farms' operational phase in the Gobi Desert, China. Nearby wind towers in desert areas, plants are less stressed, plant communities are denser, and have higher coverage than areas not occupied by wind towers. The positive effect might be due to the edge effect of areas cleared of vegetation, as well as the fact that rainwater flows along the roads and the ditches of underground power lines providing more water for vegetation, making it more diverse, more productive, and more stable (*Xu et al., 2019*).

Owing to the multiplicity of continental and marine climatic influences, the Dobrogea region (SE Romania) is the most diverse area of the Steppic European Biogeographical Region (*Rozylowicz et al., 2019*). Dobrogea is regarded as a biodiversity hotspot, with a high diversity of species protected at the national or European level (*Sârbu et al., 2006*; *Georgescu et al., 2015*; *Miu et al., 2018*; *Miu et al., 2020*). Over 63% (i.e., 9,700 km$^2$) of the Dobrogea region is part of the European Natura 2000 network, either as Special Protection Areas (SPAs, established under the Birds Directive) or Sites of Community Importance (SCIs, established under the Habitats Directive) (*Miu et al., 2018*). Despite the very good coverage of protected areas, the Dobrogea region is impacted by various factors such as quarries, wind energy farms and solar energy facilities, development of transport infrastructure, logging, invasive or non-native species, overgrazing, and land transformation for agriculture (*Petrescu, 2016*). The high wind energy potential of the Dobrogea region (*Dragomir et al., 2016*), as well as substantial European Union subsidies for renewable energies, supported the construction of several wind energy farms, outside and within the protected natural areas (*Doba et al., 2016*). A total of 890 wind towers have been installed in Dobrogea, among which 142 are within Natura 2000 sites (*Doba et al., 2016*).

Even though Dobrogea has a considerable amount of wind towers in and out of protected areas, few published studies have investigated their impact on biodiversity. For instance, *Măntoiu et al. (2020)* demonstrated that even a minor wind energy farm of 20 wind towers could trigger high bat mortality in the absence of adequate conservation measures. To the best of our knowledge, the only study investigating the impact of wind energy towers on plant species biodiversity in Romania concluded that in the Mehedinti Mountains (SW Romania) there are no significant differences between grassland communities nearby wind towers and those situated 300 m away from towers (*Pătru-Stupariu et al., 2019*). However, the study was performed on a small-scale wind farm and did not capture plant communities from wind tower technological platforms.

To contribute to a better understanding of the local-scale impact of wind energy farms on plant species susceptible to population size reductions and extinction, we analyzed the impact of wind energy farms on rare, endemic, and threatened plant species belonging to steppic grassland communities from the Natura 2000 site, Dealurile Agighiolului (Dobrogea, Romania), by contrasting the species diversity of disturbed and undisturbed plots near wind towers on Agighiol wind farm complex. We tested the hypothesis that disturbed areas (technological platforms) close to wind towers have a lower rare, endemic, and threatened plant species diversity than nearby undisturbed areas, indicating a significant local-scale impact of wind energy farms during the operational phase.
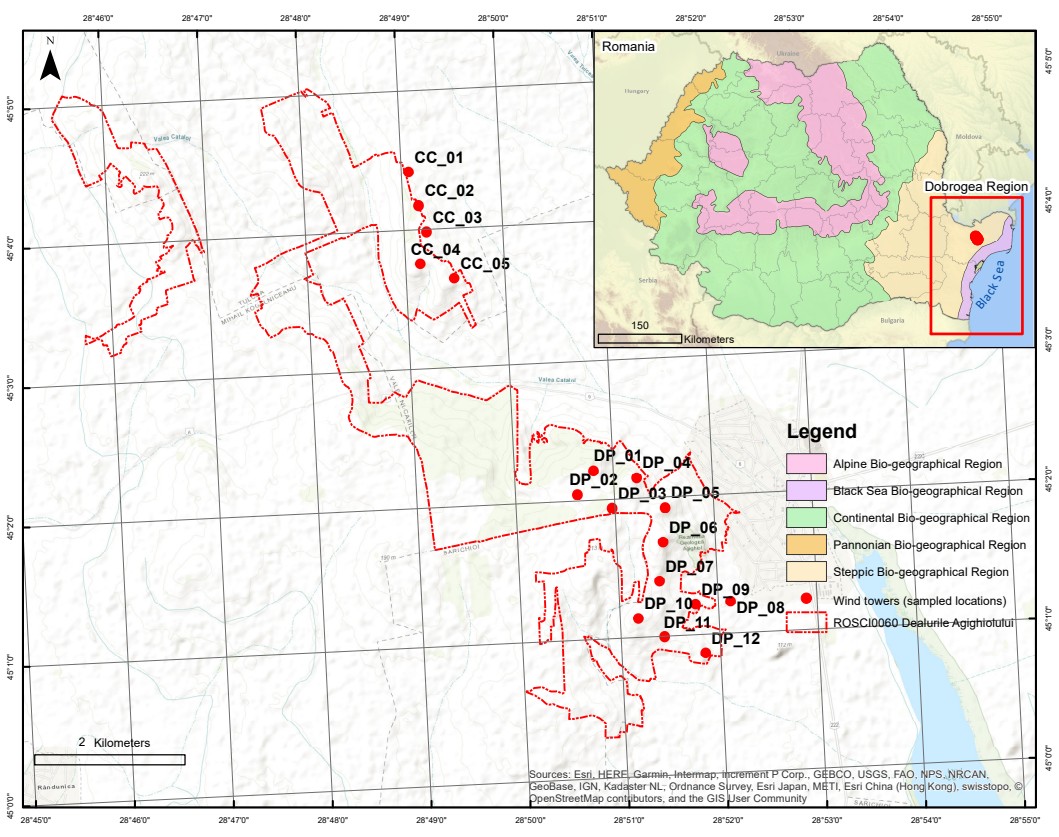

**Figure 1** **Location of Agighiol wind energy farm within Dealurile Agighiolului Natura 2000 site.** There are 17 wind towers, 12 on Pietros Hill (DP_01 to DP_012) and 5 on Caraconstantin Hill (CC_01 to CC_05).

## MATERIALS & METHODS

### Study area

Dobrogea is a geographically distinct region of 15,500 km² located in Romania's southeastern part, between the Danube River and the Black Sea (Fig. 1). The climate is humid continental with hot-summers (*Rey et al., 2007*). Long-term climatic data indicate a mean annual temperature between 10.1 and 11.8 °C and annual rainfall between 257.5 and 535.0 mm (*Bandoc & Prăvălie, 2015*).

Dobrogea's vegetation includes over 35% of the total number of syntaxa (i.e., vegetation units) identified in Romania (*Sanda & Arcuș, 1999*) and is the only Romanian region where steppe vegetation (Ponto-Sarmatic steppes) is still present (*Petrescu, 2007*). Steppes are rare in European Union countries and include endemic plant associations, such as *Agropyro-Thymetum zygioidi, Pimpinello-Thymion zygioidi,* and, *Gymnospermio altaicae-Celtetum glabratae.* Furthermore, Dobrogea hosts forest-steppe vegetation (grassland interspersed with areas of *Quercus pubescens, Carpinus orientalis,* and *Fraxinus ornus*), coastal and halophilous vegetation (mainly on the Black Sea coast and Danube Delta area), as well as reed swamps (*Scirpo-Phragmitetum* and *Thyphetum angustifoliae* associations) and
meadows in the Danube Delta (where the *Salix alba* and *Populus alba* streams dominate) (*Petrescu, 2007*).

The Natura 2000 site Dealurile Agighiolului (hereafter Dealurile Agighiolului) is located in the northeastern part of Dobrogea, Romania (Fig. 1), in a hilly area covered by steppic grassland and oak woodland surrounded by agricultural land (*Brînzan, 2013*). Dealurile Agighiolului was designated as a Site of Community Importance (Natura 2000 code ROSCI0060) in 2007, with an area of 1,433 ha. The site was designated for the conservation of three habitat types listed as having priority for conservation under Annex I of the Habitats Directive: Ponto-Sarmatic steppes (Natura 2000 code 62C0*), Ponto-Sarmatic deciduous thickets (Natura 2000 code 40C0*), and Eastern white oak woods (Natura 2000 code 91AA*). These priority habitats encompass plant communities mainly distributed in the Dobrogea region, such as *Koelerio lobatae-Thymetum zygioides* and *Paeonio peregrinae-Carpinetum orientalis* (*Chifu & Tupu, 2009*; *Tupu, 2009*; *Brînzan, 2013*). These communities harbor a significant number of protected plant species, such as the endemic *Campanula romanica*, restricted to Dobrogea, or regional species such as *Alyssum caliacrae*, *Caragana frutex*, *Centaurea kanitziana*, *Convolvulus lineatus*, *Dianthus nardiformis*, *Hedysarum grandiflorum* subsp. *grandiflorum*, *Hornungia petraea*, *Paeonia peregrina*, *Paronychia cephalotes*, *Scorzonera mollis*, *Stachys angustifolia*, *Thymus zygioides*, and *Vincetoxicum fuscatum* (*Sârbu et al., 2007*; *Dihoru & Negrean, 2009*; *Tupu, 2010*; *Petrescu, 2018*).

Concerning the number of wind towers, the Agighiol wind farm is the second largest in a protected area in Dobrogea. The wind energy farm includes 17 Gamesa G87 2 MW wind towers, of which five are in the northeast CC_01 to CC_05 and 12 in the southeast DP_01 to DP_12 (Fig. 1). The area occupied by the tower pads and access roads is 5.45 hectares (*Ministerul Mediului & Apelor i Padurilor, 2016*).

All towers are built on priority habitat 62C0* Ponto-Sarmatic steppes, except for tower DP_03, which is surrounded by habitat 91AA* Eastern white oak woods (Table 1, Fig. S1). Very close to towers CC_04 and DP_04, there are fragments of habitat 91AA*. The 62C0* habitat varies in terms of vegetation structure in the area of the wind farm. In areas where the influence of anthropogenic activity is low, there are plant associations typical for primary (natural) grasslands (e.g., *Koelerio lobatae - Thymetum zygioides*, *Stipetum lessingianae*). In intensely grazed areas or deforested areas, there are plant associations typical for secondary (semi-natural) grasslands (*Artemisio austriacae-Poëtum bulbosae*, *Taraxaco serotini-Bothriochloëtum ischaemi*) (*Tupu, 2010*; *Petrescu, 2018*).

### Data collection

Between 2015 and 2019, we surveyed rare, endemic, and threatened plant communities around the 17 towers of the Agighiol wind energy farm. We performed six visits annually per plot from March to October. The surveyed area around each towers was within a radius of approximately 50 m and included (1) a disturbed area, overlapping the technological platform, where the vegetation was removed before construction (disturbed plots of 2,500 m$^2$ each), and (2) the undisturbed area, adjacent to the disturbed one, without interventions

**Table 1  Habitats of Community interest around the wind turbines located in Dealurile Agighiolului Natura 2000 site.**

| Wind energy turbines | Habitats |
| --- | --- |
| CC_01, CC_05, DP_05, DP_06, DP_07, DP_08, DP_09, DP_10, DP_11, DP_12 | 62C0* *Ponto-Sarmatic steppes* represented by primary steppe grasslands with floristic elements characteristics of the association *Koeleria lobata - Thymetum zygioidi*, such as *Thymus zygioides, Agropyron ponticum, Allium rotundum, Koeleria lobata, Festuca valesiaca.* |
| CC_04, DP_04 | 62C0* *Ponto-Sarmatic steppes* represented by primary steppe grasslands with floristic elements characteristics of the association *Koeleria lobata - Thymetum zygioidi* and elements of the association *Paeonio peregrinae –Carpinetum orientalis* that is characteristic of habitat 91AA* Eastern white oak woods. |
| CC_02, CC_03, DP_01, DP_02 | 62C0* *Ponto-Sarmatic steppes* represented by secondary grasslands with floristic elements of the associations *Artemisio austriacae - Poëtum bulbosae* and *Taraxaco serotini-Bothriochloëtum ischaemi*: *Taraxacum serotinum, Artemisia austriaca, Bothriochloa ischaemum, Cynodon dactylon, Bromus commutatus, Bromus hordeaceus, Daucus carota* subsp. *carota, Marrubium peregrinum.* |
| DP_03 | 91AA* *Eastern white oak woods* represented by the association *Paeonio peregrinae –Carpinetum orientalis.* |

during the installation of wind towers (undisturbed plots of 2,500 m$^2$ each) (Fig. S1). The area of 2,500 m$^2$ was chosen to fit within a technological platform.

Romanian legislation requires that all taxa included in national Red Lists should be protected (*Guvernul României, 2007*). Hence, surveyed plant taxa are considered protected if they are listed as rare, endemic, or threatened in one of the following European or national regulations: Habitats Directives (*Council of the European Communities, 1992*) on the conservation of natural habitats and of wild fauna and flora, National Red Book (*Dihoru & Negrean, 2009*) or the National Red Lists (*Oltean et al., 1994*; *Dihoru & Dihoru, 1994*; *Boşcaiu, Coldea & Horeanu, 1994*).

The taxonomy considered in this paper follows the Euro+Med PlantBase (http://ww2.bgbm.org/EuroPlusMed/, accessed on 9/07/2020) and (*Sârbu, Ştefan & Oprea, 2013*). The life forms, life span, and ecological indices are according to *Sârbu, Ştefan & Oprea (2013)*. Data on vegetative reproduction and seed dispersal are according to *Dihoru & Negrean (2009)*.

## Analyses

Data on protected plant taxa inventoried in the 34 investigated plots were stored for analyses in a species by site incidence matrix (presence/absence) (Data S1). To evaluate protected plant species diversity in disturbed and undisturbed plots by wind energy farm facilities, we calculated the species richness and compared the two investigated treatments using the Wilcoxon non-parametric test (*Zar, 2010*). Because the observed species richness is overly sensitive to sampling size and effort (*Chao et al., 2014*), we calculated the estimated species richness using sample-size-based rarefaction and extrapolation with Hill numbers

($q = 0$). The analysis was performed using the iNext R package (*Hsieh, Ma & Chao, 2016*). Hill numbers represent the effective number of species calculated as the number of equally abundant species necessary to produce the observed value of the analyzed diversity (*Chao et al., 2014*). By extrapolating the species richness to double or triple the minimum observed sample sizes, the analysis may show the difference between the two environments in terms of species richness (i.e., the magnitude of the differences in richness among communities) even when sample completeness is low (*Hsieh, Ma & Chao, 2016*).

To analyze how similar/dissimilar the investigated plots are, we used Non-metric Multidimensional Scaling (NMDS) of the Jaccard dissimilarity index in the vegan R package (*Oksanen et al., 2019*; *R Core Team, 2020*). The approach produces a rank-based ordination of pairwise dissimilarity between sites and between species (i.e., objects) in a low-dimensional space (e.g., two dimensions). Two objects are more similar if they are ordinated closer together (e.g., two neighboring sites have more species in common). The number of plotted dimensions was determined using ordination stress. An ordination stress score lower than 0.2 indicates a good fit of the ordination plot, while a value close to zero indicates an outlier (site with an entirely different set of species). The ranks of the Jaccard dissimilarity distances were compared using analysis of similarity (ANOSIM) (*Oksanen et al., 2019*). We compared the mean of ranked Jaccard dissimilarities between disturbed and undisturbed sites with the mean of ranked dissimilarities within disturbed and undisturbed sites. If the ANOSIM statistic ($R$-value) is close to 1 (maximum), there is a high dissimilarity between the two groups, whereas a value close to $-1$ (minimum) indicates higher dissimilarities within groups than between disturbed and undisturbed plots (*Oksanen et al., 2019*).

Furthermore, we analyzed trait diversity in the two compared environments utilizing the total functional diversity metric available in the HillR package (*Li, 2020*). We used the following relevant traits for the inventoried plant taxa: life span (annual, perennial), life form (therophytes, hemicryptophytes, geophytes, and chamaephytes), vegetative reproduction (yes or no), and seed dispersal (barochory, anemochory, zoochory, autochory, multiple) (*Sârbu, Ștefan & Oprea, 2013*; *Dihoru & Negrean, 2009*). Total functional diversity represents the effective total distance between species of the analyzed environment and is similar to Functional Attribute Diversity when using presence/absence data (*Chiu & Chao, 2014*).

To investigate the biological processes contributing to the dissimilarity of the two compared environments, replacement of species from one plot to another, or nested species losses, we calculated beta diversity as Jaccard dissimilarity of all pairs of sites using the beta-pair function of betapart in the R package (*Baselga et al., 2020*). The beta-pair function provides a monotonic standardization of beta diversity (total dissimilarity) and two additive components: spatial turnover dissimilarity (induced by replacement of species from one plot to another) and nestedness-resultant dissimilarity (induced by nested species losses) (*Baselga & Leprieur, 2015*). The pairwise dissimilarities for each fraction of beta-diversity were tested for differences between and within environments using Kruskal–Wallis non-parametric tests (*Zar, 2010*). We tested for differences in Jaccard beta dissimilarities between the following groups: within undisturbed plots (how dissimilar are

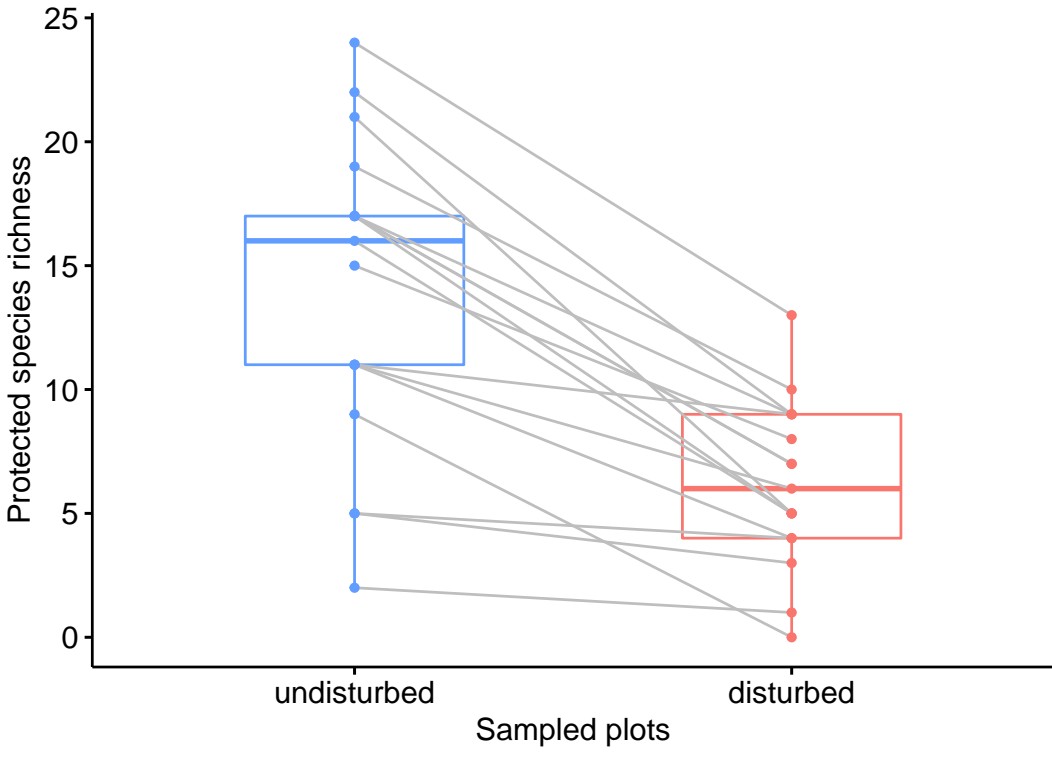

**Figure 2** **Pairwise comparison of observed species richness in undisturbed and disturbed plots from Agighiol wind energy farm.**

undisturbed plots to each other), within disturbed plots (how dissimilar are disturbed plots to each other), and between undisturbed and disturbed plots (how dissimilar are disturbed plots when compared with undisturbed plots).

Graphs and statistics, other than beta diversity, functional diversity, NDSM, rarefaction, and extrapolation, were produced using base and ggpubr R packages (*R Core Team, 2020*; *Kassambara, 2020*). Maps were produced in ArcMap 10.4 (*ESRI Redlands, CA, USA*).

## RESULTS

We identified a total of 365 plant species around wind energy towers, of which 59 were protected plant species (rare, endemic, or threatened). The number of protected species in the disturbed plots was significantly lower than in the undisturbed plots (Wilcoxon rank-sum test $W = 42.5$, $p < 0.001$), i.e., 57 species in disturbed plots and 24 species in disturbed plots (Data S1). Moreover, the rank of plots in the disturbed areas by species richness was different from the rank of paired plots in the undisturbed area (Wilcoxon signed-rank test $V = 153$, $p < 0.001$, Fig. 2).

Sample-size-based rarefaction and extrapolation with Hill numbers parameterized by species richness showed that the selected sampling units adequately captured the species diversity only in disturbed plots (Fig. 3, Fig. S2). For example, if doubling the number of plots in both treatments (e.g., from 17 to 34), the extrapolated species richness of the

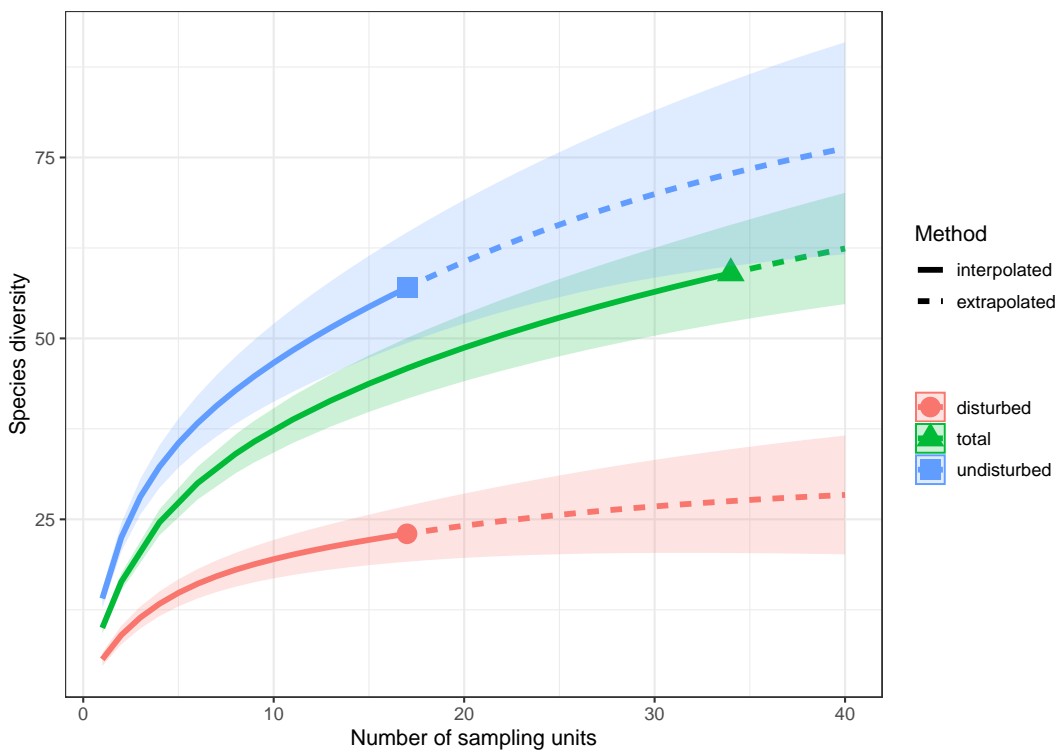

**Figure 3** Sample-size-based rarefaction and extrapolation with Hill numbers parametrized by species diversity in undisturbed, disturbed and all surveyed plots from Agighiol wind energy farm.

disturbed areas would be 29.41 (CI 95% [22.04–36.82]), a value close to the observed species richness, while the extrapolated species richness of the undisturbed area would be 72.77 (CI95% [60.11–85.43]), much higher than the observed species richness. Thus, the 17 surveyed plots in each treatment captured 93.10% (CI95% [87.70–97.20]) of the estimated number of species in disturbed plots and only 91.10% (CI95% [88.10–94.20]) in undisturbed plots. If data are pooled together (total in Fig. 3), the rarefaction and extrapolation analysis indicate sample completeness of 94.00% (CI95% [91.70–96.20]).

The most frequent plant species were *Alyssum hirsutum* and *Thymus zygioides,* present in 14 disturbed plots and 13 undisturbed plots (Fig. 4). Well represented in undisturbed plots, but to a lesser extent in disturbed plots, were *Muscari neglectum* (10 undisturbed/ 9 disturbed), *Koeleria lobata* (11 undisturbed/7 disturbed), *Tanacetum millefolium* (11 undisturbed/7 disturbed), *Bombycilaena erecta* (10 undisturbed/6 disturbed), *Echinops ritro* subsp. *ruthenicus* (9 undisturbed/7 disturbed), *Dianthus nardiformis* (10 undisturbed/ 3 disturbed), *Crocus danubensis* (9 undisturbed/ 0 disturbed), and *Scorzonera mollis* (9 undisturbed/ 0 disturbed) (Fig. 4, Data S1). Three species had two occurrences, each being found in both types of plots (*Potentilla astracanica, Scutellaria orientalis* var. *pinnatifida, Seseli rigidum* subsp. *peucedanifolium*). A total of 21 plants had only one occurrence. Of these, 19 were observed in undisturbed plots and two in disturbed plots, i.e., *Allium guttatum, Allium saxatile, Astragalus ponticus, Calepina irregularis, Campanula*

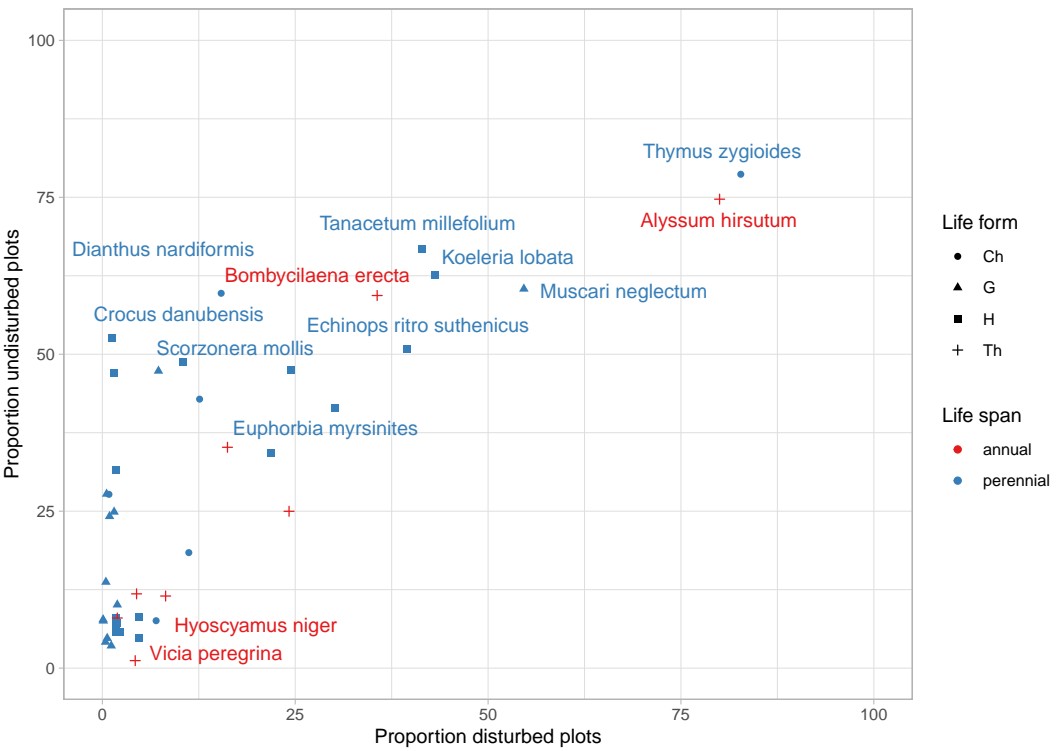

**Figure 4** **Frequency of rare, endemic and threatened plant species within undisturbed and disturbed plots from Agighiol wind energy farm.** Species names are provided for first 10 taxa as incidence in disturbed and undisturbed plots; Ch, Chamaephyte; G, Geophyte; H, Hemicryptophyte; Th, Therophyte.

romanica, *Carex brevicollis, Crocus pallasii, Cyanus thirkei, Daucus guttatus* subsp. *zahariadii, Hedysarum grandiflorum* subsp. *grandiflorum, Himantoglossum calcaratum* subsp. *jankae, Hornungia petraea, Iris sintenisii, Lathyrus cicera, Onobrychis gracilis, Orchis simia, Ornithogalum amphibolum, Piptatherum virescens,* and *Valerianella coronata,* in undisturbed plots, and *Hyoscyamus niger* and *Vicia peregrina,* in disturbed plots (Data S1).

Following the same pattern as species richness, total functional diversity was significantly lower in disturbed plots than in undisturbed plots (Wilcoxon rank-sum test $W = 221$, $p = 0.002$, Fig. S2).

Perennial plants dominated undisturbed plots and, to a lesser degree, disturbed plots (Fig. 5A). Protected perennial plants in undisturbed plots were mostly hemicryptophytes and geophytes (over 20 and 18 taxa, respectively), while chamaephytes were less represented (6 taxa) (Fig. 5A, Data S1). Among perennial plants in disturbed plots, the most represented were hemicryptophytes (nine taxa), followed by chamaephytes (five taxa) and geophytes (two taxa) (Fig. 5A, Data S1).

Of the total of recorded protected species, 28 (47.45%) were characterized by vegetative reproduction, and only seven of them were also in the disturbed plots: *Thymus zygioides, Muscari neglectum, Koeleria lobata, Tanacetum millefolium, Euphorbia myrsinites, Euphorbia nicaeensis* subsp. *dobrogensis,* and *Hyacinthella leucophaea* (Fig. 5C, Data S1).

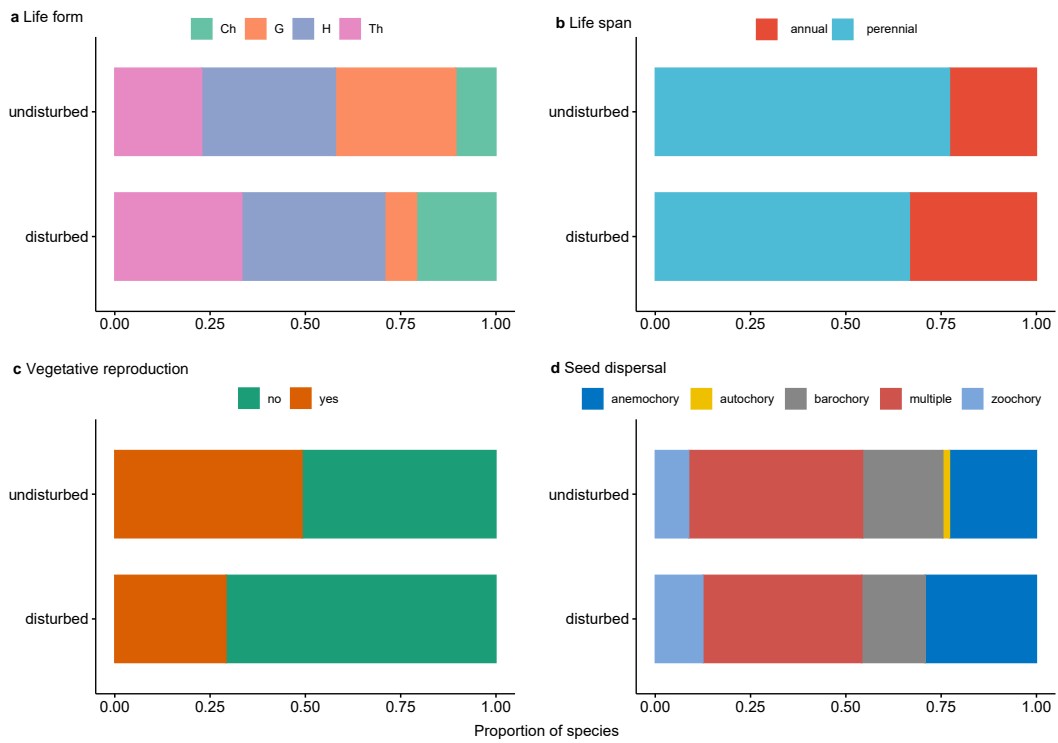

**Figure 5** **Proportion of traits for rare, endemic and threatened plant species within undisturbed and disturbed plots from Agighiol wind energy farm.** (A) Life form; (B) Life span; (C) Vegetative reproduction; (D) Seed dispersal.

Analysis of seed dispersal traits showed a similar pattern in the investigated plots (Fig. 5D, Data S1). The most represented seed dispersal systems in both environments were multiple dispersal traits (e.g., barochory and zoochory, anemochory and zoochory, autochory and zoochory, and autochory and anemochory) and anemochory. Barochory and zoochory were less represented in disturbed plots, while zoochory is less represented in undisturbed plots. Autochory is rarely seen in species present in undisturbed plots and absent in disturbed plots (Fig. 5D).

The NMDS ordination of the investigated communities had a stress value of 0.174. The first two dimensions of the ordination plots closely represent the community data. The NMDS ordination plot indicated a clear differentiation between disturbed and undisturbed plots (Fig. 6, Fig. S3). The ANOSIM test also indicated a significantly higher dissimilarity between environments than within environments ($R = 0.25$, $p < 0.001$). Following NMDS ordination, most species had high specificity for one or the other two plot types; however, some species had a weak link with either one of these environments. These were mostly species observed only in one or two plots (e.g., in Fig. 6, SP52 = *Hyoscyamus niger*, SP33 = *Bupleurum apiculatum*, SP35 = *Galanthus plicatus*, SP47 = *Cyanus thirkei*, SP39 = *Seseli rigidum* subsp. *peucedanifolium*, SP40 = *Allium guttatum*, SP44 = *Campanula romanica*, SP59 = *Valerianella coronata*, SP43 = *Calepina irregularis*, SP45 = *Carex brevicollis*, SP50
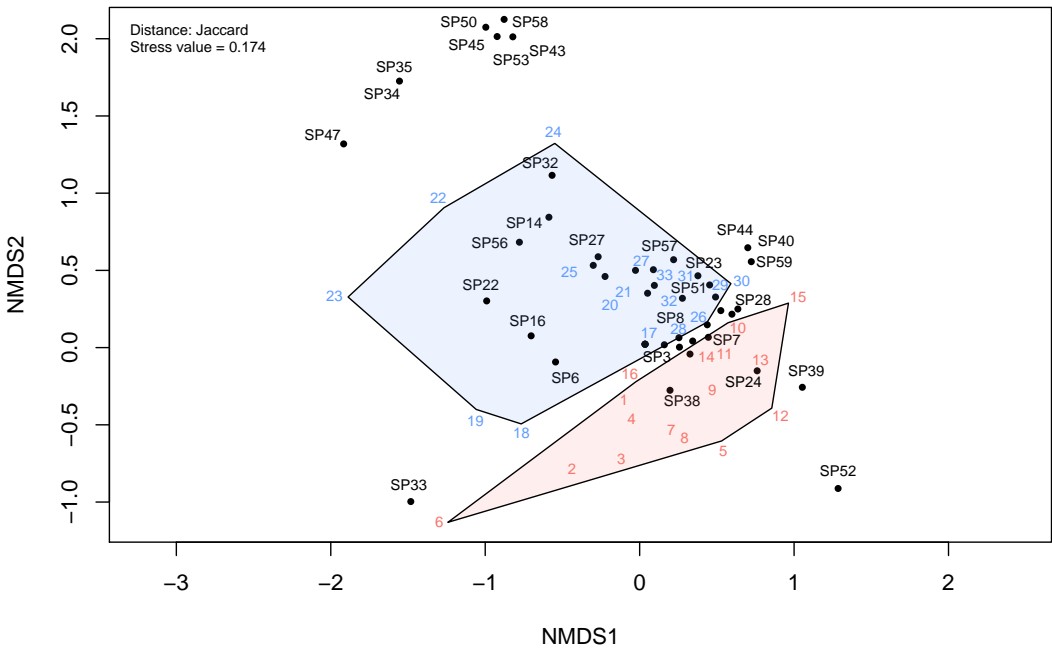

**Figure 6** **The NMDS ordination plot showing the differentiation of disturbed and undisturbed protected plant species communities from Agighiol wind energy farm.** Numbers in colors = sites; SP = species; blue = undisturbed sites; red = disturbed sites. IDs of species and sites are in Data S1.

= *Himantoglossum calcaratum* subsp. *jankae*, SP53 = *Iris sintenisii*, SP58 = *Piptatherum virescens*).

Beta diversity expressed as Jaccard dissimilarity for pairs of plots showed differences in the three compared situations: within undisturbed plots, within disturbed plots, and between undisturbed and disturbed plots (Kruskal Wallis rank-sum test = 21.92, $df = 2$, $p < 0.001$). Beta dissimilarity was significantly lower in disturbed plots than in the other two assemblages (Fig. 7A). The turnover component of beta diversity was similar in the three compared settings (Fig. 7B), while the nestedness-resultant was comparable in pairs of undisturbed and disturbed plots and higher when contrasting dissimilarity of undisturbed with disturbed plots (Fig. 7C). However, several pairs of disturbed and undisturbed plots had considerable nestedness-resultant dissimilarity (outliers in Fig. 7C).

## DISCUSSION

Our study indicates a significant negative local-scale impact of a wind energy farm on plant species diversity. By comparing rare, endemic, and threatened plant diversity from areas disturbed by wind towers (technological platforms and access roadsides) with nearby undisturbed areas, we detected a significantly lower alpha diversity and total beta dissimilarity of disturbed areas within the Dealurile Agighiolului. This sharp contrast is documented following ten years of the Agighiol wind energy farm's operational history, suggesting that routine wind farm maintenance without enforcing conservation measures hampered the colonization of areas disturbed during the construction phase.

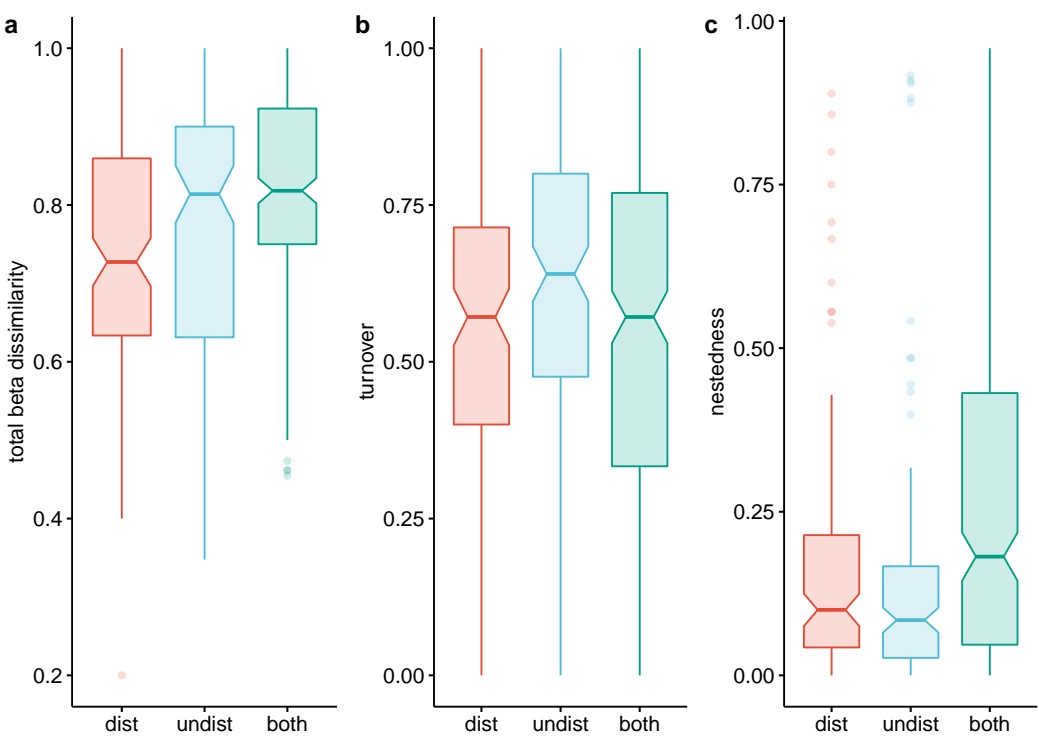

**Figure 7** **Beta-diversity expressed as Jaccard dissimilarity in disturbed, undisturbed, and all plots from Agighiol wind energy farm.** (A) Total beta dissimilarity; (B) Spatial turnover dissimilarity; (C) nestedness-resultant dissimilarity.

The number of protected species recorded in the investigated area was three times higher than in the environmental impact assessment carried out prior to the construction of the wind farm (59 taxa versus 18 taxa) (*Badea et al., 2012*), suggesting that the ex-ante impact assessment failed to provide high-quality data to environmental authorities and wind farm operators. The lack of accuracy of environmental studies is a common issue (*Nita et al., 2015*), including when analyzing plant communities (*Fraga et al., 2008*; *Silva & Passos, 2017*).

Our extensive survey of plant communities surrounding the wind energy towers within Dealurile Agighioului indicates that less than 40% of the total inventoried protected plant species colonized the disturbed sites. The lower species richness observed in disturbed sites was also evident when compared to the undisturbed plots. None of the disturbed sites had a higher species alpha diversity than nearby undisturbed surveyed plots (Fig. 2). Most undisturbed plots hosted a much higher number of protected plant species than the disturbed plots. We found smaller differences between the number of species of disturbed and undisturbed sites for very few pairs of sites (CC_02, CC_03, and DP_07 wind towers). For wind energy towers located in secondary grassland, the number of inventoried protected species was low in the disturbed and undisturbed plots (i.e., CC_2, CC_3, DP_2), varying between 1 and 5 species. For wind energy towers located in primary grassland, the number of protected species was high (11 to 24 species). These results suggest a correlation between

the habitat surrounding each wind tower and the number of species from the adjacent disturbed area, plant communities near wind towers representing a pool of species for the impacted areas. However, the disturbance role of the wind farm in our area is evident considering that the survey of disturbed habitats was almost complete (observed number of species = 24, extrapolated number of species = 29, Fig. 2).

Sample-size-based rarefaction and extrapolation with Hill numbers indicated an accurate estimation of species richness in disturbed habitats (Fig. 3), supporting the conclusion that recovery after wind farm construction is far from complete even after ten years of maintenance activities. Of the 57 plant taxa present in undisturbed areas, only 22 taxa were also found in disturbed areas (e.g., *Adonis flammea, Alyssum caliacrae, Alyssum hirsutum, Astragalus glaucus, Bombycilaena erecta, Centaurea kanitziana, Cerastium gracile, Dianthus nardiformis, Echinops ritro* subsp. *ruthenicus, Euphorbia myrsinites, Euphorbia nicaeensis* subsp. *dobrogensis, Hyacinthella leucophaea, Koeleria lobata, Minuartia adenotricha, Minuartia hybrida, Muscari neglectum, Potentilla astracanica, Scutellaria orientalis* var. *pinnatifida, Seseli rigidum* subsp. *peucedanifolium, Silene supina, Tanacetum millefolium, Thymus zygioides*).

Trait diversity was significantly lower in disturbed areas than nearby undisturbed areas, suggesting that few traits may favor colonization of these impacted plots. As shown, disturbance constrained the colonization of most nearby species; however, the disturbance may favor the development of species well adapted to the new environmental conditions (*Patykowski, 2018*; *Herben, Klimešová & Chytrý, 2018*). In our study, the winner species were *Alyssum hirsutum*, a rare annual species in Romania (*Oprea, 2005*), and *Thymus zygioides*, a perennial with vegetative and generative reproduction, also rare in Romania (*Oltean et al., 1994*). Both species are well represented in the Ponto-Sarmatic steppes of Dobrogea. Analysis of the life span of protected plants from the two environments showed that the proportion of annual plants was slightly higher in disturbed plots than in undisturbed plots (Fig. 5B). *Herben, Klimešová & Chytrý (2018)* suggest that annual plants are well adapted to various degrees of disturbance severity and frequency and may flourish in the first stages of disturbance, but over time are replaced by perennials. For example, annual plants might be favored by the anemochory trait, which is the case with *Alyssum hirsutum* inventoried in 27 plots, and *Bombycilaena erecta* inventoried in 16 plots. In our case, the relatively higher proportion of plants in the disturbed plots may suggest repeated disturbance events during the ten years of the Agighiol wind farm's operational phase. Disturbance can also favor species with particular traits, such as growth on stony substrates (*Schnoor & Olsson, 2010*; *Kompała-Bąba et al., 2019*). This is observed in *Alyssum caliacrae*, a species representative of Dobrogea's stony grasslands (*Dihoru & Negrean, 2009*), which found better propagation conditions in disturbed areas and maintained a higher abundance compared with the adjacent undisturbed habitat (*Urziceanu, Anastasiu & Șesan, 2020*).

The dissimilarity of undisturbed and disturbed plots is also evident from the NDSM ordination (Fig. 6) and beta-diversity analysis (Fig. 7). The spatial turnover component of beta-dissimilarity was higher than nestedness-resultant in all compared environments (within undisturbed plots, within disturbed plots, and between undisturbed and disturbed plots) due to the presence of many species restricted to an environment (*Baselga &*
*Leprieur, 2015*). Furthermore, the lower nestedness of both environments indicates a significant heterogeneity of disturbed and undisturbed communities, with fewer than expected species in common in each assemblage (e.g., *Thymus zygioides*, *Alyssum hirsutum*). The higher turnover and lower nestedness observed for the undisturbed plots can be explained by plot location within Dealurile Agighiolului, specifically by the nearby habitats. Around the towers located on primary steppe grasslands (towers: CC_01, CC_04, CC_05, DP_04, DP_05, DP_06, DP_07, DP_08, DP_09, DP_10, DP_11, and DP_12), we inventoried plants species characteristic for this habitat, such as *Adonis vernalis, Allium guttatum, Allium saxatile, Alyssum caliacrae, Campanula romanica, Crocus pallasii, Dianthus nardiformis, Echinops ritro* subsp. *ruthenicus, Euphorbia myrsinites, Hedysarum grandiflorum* subsp. *grandiflorum, Hornungia petraea, Koeleria lobata, Minuartia hybrida, Ornithogalum amphibolum, Ornithogalum sigmoideum, Paronychia cephalotes, Salvia nutans, Scutellaria orientalis* var. *pinnatifida, Seseli rigidum* subsp. *peucedanifolium, Silene supina, Stachys angustifolia, Sternbergia colchiciflora, Tanacetum millefolium, Trigonella gladiata, Valerianella coronata,* and *Vincetoxicum fuscatum*. In contrast, around the towers located in ruderalized secondary grasslands (towers: CC_02, CC_03, DP_01, and DP_02), we observed fewer species, but also a different set of species (*Adonis flammea, Alyssum hirsutum, Bombycilaena erecta, Bupleurum apiculatum, Colchicum triphyllum, Muscari neglectum,* and *Scorzonera mollis*). Ruderalization is due to intensive grazing and agricultural activity, which interfere with the wind energy farm area.

The present study provides data suggesting a clear local-scale impact of wind energy farms on plant species prone to population size reductions and extinction. Plant communities were inventoried annually between 2015 and 2019, thus sufficiently capturing plant communities at disturbed sites. Despite our approach's robustness, the lack of rigorous surveys before the construction of the wind energy farm (*Badea et al., 2012*) limits the study to a post hoc one. Thus, some of the differences might be due to initial disturbance and not due to the operational phase. Furthermore, because we focused on a single wind energy complex in an area with many plant communities endemic to Romania, our results may need to be validated in other environments such as wind energy complexes within habitats with more common species.

## CONCLUSIONS

Although the number of wind energy farms within protected areas has significantly increased in the past decade, few studies have quantified their impact on protected plant species recovery. Our study suggests that wind energy farms' operation affects the local diversity of protected plant species. Disturbed areas have a significantly lower recovery rate in the absence of restoration and post-restoration measures at the technological platforms. The high number of protected species in the undisturbed plots indicates that in the nearby areas of wind towers the plant communities are not affected by construction and operation activities. We conclude that after ten years of operating the Agighiol wind farm, the effects on protected plant communities are present only in areas where the vegetation has been removed, and maintenance activities are carried out regularly. Hence, enforcing

conservation activities, such as a narrower area available for maintenance and operation of towers, may allow the expansion of the populations of protected plants and a higher rate of colonization from nearby sources.

## ACKNOWLEDGEMENTS

We would like to thank Enel Green Power Romania and the custodian of the Dealurile Agighiolului Natura 2000 protected area for allowing the research activity in the wind energy farm Agighiol. We also thank the two reviewers for their extensive comments and suggestions and Edward F. Rozylowicz for proofreading and suggestions, which helped us improve the manuscript.

### Funding

Laurentiu Rozylowicz was supported by a grant of Executive Agency for Higher Education, Research, Development and Innovation Funding (PN-III-P1-1.1-TE-2019-1039). The funders had no role in study design, data collection and analysis, decision to publish, or preparation of the manuscript.

### Grant Disclosures

The following grant information was disclosed by the authors:
Executive Agency for Higher Education, Research, Development and Innovation Funding: PN-III-P1-1.1-TE-2019-1039.

### Competing Interests

The authors declare there are no competing interests.

### Author Contributions

- Mihaela Urziceanu and Paulina Anastasiu conceived and designed the experiments, performed the experiments, analyzed the data, authored or reviewed drafts of the paper, and approved the final draft.
- Laurentiu Rozylowicz analyzed the data, prepared figures and/or tables, authored or reviewed drafts of the paper, and approved the final draft.
- Tatiana Eugenia Sesan analyzed the data, authored or reviewed drafts of the paper, and approved the final draft.

### Field Study Permissions

The following information was supplied relating to field study approvals (i.e., approving body and any reference numbers):

The Enel Green Power Romania (#600189) and the custodian of the Dealurile Agighiolului Natura 2000 protected area allowed the research activity in the wind energy farm Agighiol.

## Data Availability

R script and data to reproduce analyses and figures in the article are available in the Supplementary Files.

## Supplemental Information

Supplemental information for this article can be found online at http://dx.doi.org/10.7717/peerj.11390#supplemental-information.

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
