# Peer review of "Local-scale impact of wind energy farms on rare, endemic, and threatened plant species"

_PeerJ, doi:10.7717/peerj.11390_

## Round 0.1 · original submission · Major Revisions

Reviewers were complimentary of your research but had concerns about the presentation in terms of discussion, flow, and language. Of note is the difficulty in evaluating a true disturbance level given the lack of description of community structure prior to installation of the wind farm. Other concerns are indicated via review comments as well as in the attached pdf. Finally, I suggest acquiring the help of a fluent English speaker for simplifying and improving the language and phrasing.

Reviewer 1 ·

Basic reporting

The manuscript “Habitat disturbance by wind energy farms effects on protected plant diversity at local scale. A case study of Dealurile Agighiolului Natura 2000 site” by Urziceanu et al surveyed plant species around the disturbed and undisturbed plots near wind farm turbines to examine the impact of wind farm on plant biodiversity. The topic of this study is worth investigating and the findings that disturbance decreased species richness provide valuable evidence for the wind farm impacts. The manuscript is well-written. Below I have some questions about the methodology and findings.

Experimental design

I suggest to include basic geographical information about the study region, in particular climate conditions and vegetation types. This information provides important background to understand the wind farm impact on plants because climate also affects the resilience of plants to disturbance.

Regarding the methodology, the authors took samples from within 50 m distance from the wind farm. What is the justification for this distance choice? How does the wind farm disturbance impact vary with distance? Besides, the authors need to give more information about how to define undisturbed and disturbed plots in their field survey. This is very important for this study but for now, there is just a vague description that is not clear to me. It would also be great to include photos of the disturbed plots.

It seems the authors focused on the impact of protected species and found decreased diversity, which is fair. I am wondering what would be the impact on unprotected species? Because the protected species might be more vulnerable to disturbance and the unprotected species might be more resilient to disturbance (which is why they are not on the red list). A related question is about the general impact of wind farm on vegetation, in addition to plant diversity. There are also properties like vegetation coverage, biomass, etc. It would be great if the authors can put their findings of plant diversity under the context of general impacts of wind farms on vegetation, which makes the paper more attractive to a broader audience. Perhaps adding relevant discussion in the introduction or discussion section would be good enough. For example, there are also relevant studies reporting positive effects of wind farm on plants at either local and large scales, e.g., Xu 2018 and Li 2018.

Validity of the findings

Minor comments:
It would be helpful to include some photos of the landscape of the sampled wind farm region.

Where are the sampled locations? Is it possible to please add them to the map?

Figure 1. Please include latitude/longitude and the location of the study region from a global or regional perspective, e.g., a zoomed map view to indicate the location from the world/regional map.

L165-169: I feel the authors need to provide more information about the disturbed area. Is it an area previously occupied by wind farm facility and later recovered, or is it still currently occupied by the operation of wind farm facility? They are two different cases. The former denotes a legacy disturbance effect while the latter denotes a persistent disturbance effect.

L250-251: “The annuals are more frequent in disturbed plots, comparing with the perennial, more frequent in undisturbed plots”. The wind farm disturbance indeed changed plant diversity, but how to explain the differences between perennials and annuals?


References
Xu, K., He, L., Hu, H., Liu, S., Du, Y., Zhiwei, W., … Li, L. (2019). Positive ecological effects of wind farms on vegetation in China ’ s Gobi desert. Scientific Reports, (October 2018), 1–11. https://doi.org/10.1038/s41598-019-42569-0
Li, Y., Kalnay, E., Motesharrei, S., Rivas, J., Kucharski, F., Kirk-Davidoff, D., … Zeng, N. (2018). Climate model shows large-scale wind and solar farms in the Sahara increase rain and vegetation. Science, 361(6406), 1019–1022. https://doi.org/10.1126/science.aar5629

Reviewer 2 ·

Basic reporting

The authors of this manuscript, entitled ” Habitat disturbance by wind energy farms effects on protected plant diversity at local scale. A case study of Dealurile Agighiolului Natura 2000 site (Dobrogea, Romania)” present a study on the impact of wind farm developments on natural habitats, with an example from Dobrogea region (Romania). Their major goal is to evaluate the pressures on protected plant species after 10 years of operational history of the Agighiol wind energy farm (Dealurile Agighiolului Natura 2000 site, Dobrogea, Romania) by comparing across the 17 selected wind turbines the plots covering disturbed habitats with their corresponding proximal pair-plot covering undisturbed habitats, but also by partitioning beta-diversity components within each plot type and among the two types.

The language is quite complicated and wording is entangled. I suggest acquiring the help of a native speaker for simplifying and improving the language and wording/phrasing, which in the current state suffers greatly and affects reading, being in many cases unnecessarily complicated. Such simplification of the language would serve well the entire study by enhancing reading and proper delivery of information, subsequently increasing the value of the paper.

The manuscript has sufficient references to develop the background and support the findings. The figures are well-designed, however, the figure captions should be removed from the figure and be presented as text.

The structure of the manuscript is not ideal. Some paragraphs have to be moved between sections, some sections (especially the Discussion part) have to be better developed and clearly presented in relation to the obtained results.

Experimental design

The paper presents original data and the study falls within the scope of the journal.

It seems, however, that the authors have missed in presenting the working hypothesis/hypotheses, aspect that I've highlighted in my review, presented in the attached pdf. The imact of the results is well defined and meaningful, bringing more insight over the proposed research topic.

The statistical analyses are correct and fit the data used in this study. The given details are sufficient and offer the possibility to replicate them. Some additional anayses are suggested for a more thorough approach of the subject, including functional diversity, species eveness and providing additional information on plant communities.

Some methods should be better explained to highlight their importance and meaning and why where they chosen for this study.

Validity of the findings

The study is valuable by approaching an important topic in the context of increased infrastructure developments nowadays for the production of green energy, however it is poorly presented and lacks a good flow in both language and argumentation.

It seems that the disturbance level is rather difficult to evaluate in the case of undisturbed plots due to a lack of information on the community structure prior to wind farm development, which in my opinion affects the value of the results and overall conclusions. There is a way to overcome this caveat that I present in the pdf containing Major and Minor observations to the msc.

Raw data are provided and should provide enough support for any replication of the study.

Conclusions should be better developed in relation to the hypothesys that will be formulated by the authors.

These aspects can be corrected by reconsidering all sections of the manuscript, including additional analyses.

Additional comments

The suggested changes would be required before the manuscript can be considered for publication in PEERJ. I thus recommend a Major Revision and would be happy to re-visit the next version of the manuscript, when submitted, and receive the author’s reply to my comments.

I wish the authors good luck and present my Major Observations and Minor comments in the attached pdf.

Annotated reviews are not available for download in order to protect the identity of reviewers who chose to remain anonymous.

---

## Round 0.2 · accepted · Accept

I am accepting your manuscript, however, there are a couple of reviewer comments on your revised manuscript I would like you to address. 1. Please make sure the sampled locations are clear on the map. 2. There remain some unclear sentences (grammar errors) at Lines19-21 and Lines 350-353.

Reviewer 1 ·

Basic reporting

The revised manuscript addressed most of my comments. But I am not satisfied with some responses.
For example, the authors replied my original comment "Where are the sampled locations? Is it possible to please add them to the map?" by "We redraw the map and included more details." However, I could not find the sampled locations on the redrawn map.

There are still occasional unclear sentences or with grammatic errors, for example,
L19-21; L350-353

Experimental design

NA

Validity of the findings

NA

Additional comments

NA